# Multiple Facets of Nitrogen: From Atmospheric Gas to Indispensable Agricultural Input

**DOI:** 10.3390/life12081272

**Published:** 2022-08-19

**Authors:** Nkulu Rolly Kabange, So-Myeong Lee, Dongjin Shin, Ji-Yoon Lee, Youngho Kwon, Ju-Won Kang, Jin-Kyung Cha, Hyeonjin Park, Simon Alibu, Jong-Hee Lee

**Affiliations:** 1Department of Southern Area Crop Science, National Institute of Crop Science, RDA, Miryang 50424, Korea; 2National Crops Resources Research Institute (NaCRRI), NARO, Entebbe 7084, Uganda

**Keywords:** nitrogen, agriculture, environment, sustainable food production

## Abstract

Nitrogen (N) is a gas and the fifth most abundant element naturally found in the atmosphere. N’s role in agriculture and plant metabolism has been widely investigated for decades, and extensive information regarding this subject is available. However, the advent of sequencing technology and the advances in plant biotechnology, coupled with the growing interest in functional genomics-related studies and the various environmental challenges, have paved novel paths to rediscovering the fundamentals of N and its dynamics in physiological and biological processes, as well as biochemical reactions under both normal and stress conditions. This work provides a comprehensive review on multiple facets of N and N-containing compounds in plants disseminated in the literature to better appreciate N in its multiple dimensions. Here, some of the ancient but fundamental aspects of N are revived and the advances in our understanding of N in the metabolism of plants is portrayed. It is established that N is indispensable for achieving high plant productivity and fitness. However, the use of N-rich fertilizers in relatively higher amounts negatively affects the environment. Therefore, a paradigm shift is important to shape to the future use of N-rich fertilizers in crop production and their contribution to the current global greenhouse gases (GHGs) budget would help tackle current global environmental challenges toward a sustainable agriculture.

## 1. Introduction

Agriculture is one of the major components of many development programs aiming at transforming economies, and it is perceived as a core economic sector with great potential to induce sustainable economic growth in several countries [1]. Agriculture is also central to achieving essential development goals, including ensuring food security and improving human and animal nutrition, as well as achieving employment stability and social growth [2].

To sustain their growth and productivity, plants require nutrients, which are essential for successful growth and achieving optimum yields. Among them, nitrogen (N), phosphorus (P), and potassium (K), known as primary macronutrients, and sulfur (S), calcium (Ca), and magnesium, identified as secondary macronutrients, have been reported to play a preponderant role. Of all these nutrients, N has proven indispensable in several ways, and N deficiency in the soil has been shown to result in impaired growth and development that culminates in compromised productivity and quality of crops [3,4]. To date, dosages of primary macronutrient applications for various crop species have been optimized, and evidence of increased crop productivity has been recorded. One should note that the dosage of a particular fertilizer is a function of many factors, including plant nutrient requirement and growth stage, the variety of a crop species under cultivation, the type and the concentration of the fertilizer, and soil type and density. In the same way, the availability of nutrients to the plant is affected by the soil type, the pH, and the cationic exchange capacity (CEC) of the soil [5]. Likewise, abiotic factors, such as salinity, drought stress, or heavy-metal toxicity can impair the availability of nutrients to the plant, thus resulting in poor growth and productivity.

In recent years, many reports highlighted a growing interest in the use of an optimized plant nutrition, as well as the promotion of fertilizer use efficiency, which have emerged over the concerns that excessive application of mineral fertilizers, especially those rich in N, has proven to be one of the major causes of greenhouse gases (GHGs) emissions in agriculture [6,7,8]. Some voices emphasized the necessity to transition to sustainable agricultural production systems, while others sustained that the use of N-rich fertilizers is necessary for more food production to feed the growing global population. The paradigm of plant nutrition lies in the necessity to tackle the issue of soil fertility for efficient and successful crop cultivation and increased productivity, associated with an enhanced nitrogen use efficiency (NUE) by plants, while considering the sustainability and the resilience of food production systems. Recent alarming global warming records have set an imperative to reduce the emission of GHGs in all sectors. In agriculture (crop production), reducing the excessive application of N-rich fertilizers is a major target.

This review aims to provide an expanded view of the roles of N and N-containing compounds in plants, and it presents multiple facets of N under various conditions in a single location. We equally discuss the dynamics of N availability, use, remobilization, deficiency, and toxicity in soil and how these events affect the plant. This work also assesses and highlights the progress made in our understanding of N metabolism in plants, diving from the empirical background to the future use of N, from the perspective of global environmental challenges.

## 2. Basic Properties of Nitrogen and N-Containing Compounds

Nitrogen is a gas present in the air with an atomic mass of 14.007, a boiling and melting points of 77.36 K and 63.15 K, respectively, and a density of 0.0012506 g·cm^–3^. N was first discovered in 1772 by the Scottish physician and chemist Daniel Rutherford [9]. N is the fifth most abundant element in the universe after hydrogen (H, first), helium (He, second), oxygen (O, third), and neon (Ne, fourth), and it makes up approximately 78.1% of the earth’s atmosphere. Reports indicate that an estimate of 4000 trillion tons of N can be found in the atmosphere in the form of N_2_ (https://pubchem.ncbi.gov/element/Nitrogen, accessed on 31 May 2022). The most popular use of N is for the production of ammonia (NH_3_) when combined with hydrogen (H), in a process called the “Haber process” [10]. Then, large amounts of NH_3_ are used to produce mineral fertilizers in a process known as the “Ostwald process”, among other uses [11]. N is found in all living systems as part of the makeup of biological compounds. During the decomposition of organic matter (OM), sodium nitrate (NaNO_3_) and potassium nitrate (KNO_3_) are formed. Other inorganic N compounds include HNO_3_, NH_3_, the oxides (nitric oxide (NO), nitrogen dioxide (NO_2_), N_2_O_4_, and nitrous oxide (N_2_O)), and cyanides (CN). NO_2_, NO, nitrous acid (HONO), and nitric acid (HNO_3_) belong to a group of highly reactive gases known as oxides of nitrogen or nitrogen oxides (NOx) [12].

NO_2_ is formed during nitrification and denitrification processes, in which N_2_O and NO are released, with N_2_O reported to be formed from NO_3_-dependent NO formation [13]. N_2_O is a potent greenhouse gas (GHG), with a global warming potential (GWP) 300 times greater than the mass of carbon dioxide (CO_2_) in the atmosphere, right before methane (CH_4_) and CO_2_ [14,15]. As for NO, reports indicate that this molecule is versatile and bioactive, with the potential to diffuse through biological membranes owing to its physiological properties. NO can act as a signaling molecule, which may involve a very wide web with reactive oxygen species (ROS). However, excessive accumulation of NO induces stressful conditions in plants [16,17]. NO and its derived molecules are reported to be involved in abiotic and biotic stress response mechanisms.

## 3. Historical Use of Nitrogen and N-Rich Fertilizers in Agriculture

The history of agriculture revealed that a number of plant species were domesticated from wild ancestors [18,19,20,21,22,23] during the Early Pre-Pottery Neolithic period, at various locations and different times between 10,500 and 10,100 years before common era, BCE [24,25,26]. During this period, crops exhibited a low productivity, poor yields, and poor quality, mainly attributed to their genetic makeup [25,27,28,29,30,31,32]. Since then, significant progress has been made using plant breeding and the establishment of plant nutrition schemes. According to Pennazio [33], the development of mineral nutrition of plants began between the 17th and 18th centuries. The patterns of mineral fertilizer applications in different cropping systems, as well as their impact on the environment, continue to nourish the debate globally [34,35,36,37,38,39,40,41,42,43,44,45,46,47]. However, the increase in food demands due to the rapid increase in the global population has shown the necessity to enhance the productivity of food crops. To achieve that, the common strategies used are crop improvement and the use of mineral fertilizers during crop cultivation, which have increased over the years [48,49]. Despite the recorded progress in plant breeding, N remains an indispensable macronutrient, among the 14 mineral elements (macro- and micronutrients) required by plants for optimum growth and development, high productivity, quality, fitness, and resistance toward environmental and biotic stresses [50,51].

Data indicate that the application of N-rich fertilizers increased over the years concomitant with the expansion of crop cultivation areas and the use of improved or high-yielding crop varieties that are demanding of nutrients, particularly observed in large-scale farming systems. This trend could be partially attributed to the increase in food demands subsequent to the increase in the world’s population. An estimate of the global population growth shows that the number of people on Earth increased by 219.1% in 72 years, from about 2,499,322,157 in 1950 to nearly 7,975,105,156 in 2022 (https://www.macrotrends.net/countries/WLD/world/population, accessed on 15 July 2022), in contrast to the decreasing pattern of the annual population growth rate during the same period (nearly 1.75% in 1951 down to 0.83% in 2022, a decrease by 47.4%). Available data on land coverage, as reported by the Food and Agriculture Organization of the United Nations (FAO) statistics (https://www.fao.org/faostat/en/#data/LC/visualize, accessed on 15 July 2022), suggest that herbaceous crops occupied about 1,877,418.8 ha of cultivated lands in 1992 compared to 1,904,136.4 ha in 2020 globally. In addition, the report on the use of nutrients for agricultural production during the last six decades indicates that nearly 4,155,951,874 metric tons of N was used between 1961 and 2020 (11,455,804.3 mt in 1961 and 113,291,696.7 mt in 2020) (https://www.fao.org/faostat/en/#compare, accessed on 15 July 2022). When compared with other macronutrients (phosphorus (P) and potassium (K)), N is by far the most abundantly used plant nutrient element in agriculture (nearly 4.15 billion tons (BT) within 59 years). For instance, a report by FAO showed that during 1961–2020, more than 1.9 BT of phosphate (P_2_O_5_) and 1.4 BT of potash (K_2_O) were used for agricultural production globally (Figure 1).

## 4. Essentiality of Nitrogen, Sources, and Availability

Arnon and Stout [52] proposed three criteria for the essentiality of a plant nutrient as follows: (1) a deficiency of the element makes it impossible for a plant to complete its lifecycle; (2) the deficiency is specific to the element in question; (3) the element is directly involved in the nutrition of the plant, such as a constituent of an essential metabolite or required for the action of an enzyme system. Other reports suggested that a more inclusive definition of the essentiality of a plant nutrient should be considered, which is not limited to those elements when they are deficient in the soil or unavailable, when they may cause symptoms of deficiency, and when their correction may involve an external supply through fertilization [53,54,55]. Therefore, on the basis of the abovementioned criteria, N and 15 other elements, namely, carbon (C), hydrogen (H), oxygen (O), phosphorus (P), potassium (K), calcium (Ca), magnesium (Mg), sulfur (S), iron (Fe), manganese (Mn), copper (Cu), zinc (Zn), molybdenum (Mo), boron (B), and chlorine (Cl), are considered essential for the growth of higher plants. Of this list, C, H, and O are acquired by the plant directly from the air and soil water, while the remaining 13 are supplied by the soil [56,57]. On the basis of the amounts in which the essential nutrients are acquired by the plant (except C, H, and O), they are categorized as primary (N, P, and K) and secondary macronutrients (Ca, Mg, and S), and micronutrients (Fe, Mn, Zn, Cu, B, Mo, and Cl).

Plants are sessile living organisms, which do not have the ability to move from one environment to another looking for food in the case of nutrient deficiency in their immediate environment [58,59]. Soil is the principal source of nutrients (including N) necessary for plants to complete their life cycle [60,61]. The availability of N is affected by various factors that may be extrinsic or intrinsic to the plant [62,63,64,65,66,67]. Crop productivity relies heavily on N fertilization [68,69,70]. N is found abundantly in the air but in a form that plants cannot absorb. The major sources of N are atmospheric nitrogen gas N_2_, exogenous N supply, and OM through fertilization. Atmospheric N_2_ gas (plentiful in the air but cannot be absorbed by plants in this form) is acquired through the nitrogen fixation cycle mediated by plant species belonging to Fabaceae (leguminous, along with a few non-leguminous plants containing nodules in their roots) in a symbiotic association with soil microorganisms known as nitrogen-fixing bacteria belonging to the genus *Rhizobium*. The latter mediate the conversion of atmospheric N_2_ to ammonium (NH_4_) in the soil, which in turn is converted to nitrate (NO_3_) during the nitrification process. According to Maier [71], the growth of many bacteria, either free-living in the environment or in symbiosis with plants, is promoted during N fixation in areas where fixed N is deficient in the soil. The root nodule bacteria have many O_2_-binding terminal oxidases, with a high O_2_ affinity, which are associated with N_2_ fixation and help maintain a steady O_2_ supply, coupled with ATP supply for high energy-demanding N_2_ fixation. The fertility of soil depends on several factors, including the quantity and quality of nutrients present in the soil, soil physical, biological, and chemical properties [72].

NO_3_ and NH_4_ are the major forms of N taken up by the plant, with NO_3_ being the most abundant [70]. However, roughly half of N (all N sources and forms considered) is used by the plant, while the remainder is either lost to groundwater (percolation or leaching), consumed by soil microorganisms (bacteria) involved in the decomposition of OM to humus, or converted back to atmospheric N_2_ via the denitrification process.

## 5. Nitrogen-Based Fertilizers

Nitrogen is commonly applied using commercial N-containing fertilizers or OM (composts, liquid organic fertilizers, and animal feces) [73,74,75,76,77]. One of the most abundant forms of N commercially available is urea 46% N ((CO(NH_2_)_2_) or CH_4_N_2_O) also known as carbamide, or it can be found together with P in the form of diammonium phosphate (DAP, (NH_4_)_2_HPO_4_). The latter is also referred to as ammonium monohydrogen phosphate, ammonium hydrogen phosphate, or ammonium phosphate dibasic, with the typical formulation of 18% N, 46% P_2_O_5_, 0% K_2_O) or triple superphosphate (TSP) referred to as calcium dihydrogen phosphate or monocalcium phosphate ((Ca(H_2_PO_4_)_2_·H_2_O), containing 45% phosphate (P_2_O_5_) (0–45–0), 15% Ca), whereas, K is supplied as potash (K_2_O) [67]. Together, they make up the trio widely known as NPK. In the soil, N is available to the plant and transported in the form of NO_3_ and NH_4_, with NO_3_ transport being predominant over NH_4_ and the major source N [78,79]. DAP is known as the world’s most widely used phosphorus fertilizer and one of the known water-soluble ammonium phosphate salts that can be produced when ammonia (NH_3_) reacts with phosphoric acid (H_3_PO_4_) ((NH_4_)_2_HPO_4_ (s) ⇌ NH_3_ (g) + (NH_4_)H_2_PO_4_ (s)) [80]. Reports indicate that, when applied as a source of N and P, DAP temporarily increases the soil pH but becomes more acidic over the long term upon nitrification of the NH_4_, and it is said to be incompatible with alkaline chemicals due to the high potential for the NH_4_ ion to convert to NH_3_ in a high-pH environment (pH 7.5–8.0).

## 6. Multiple Roles of Nitrogen and N-Containing Compounds

Successful plant growth and development, reproduction, and productivity are the result of complex processes, including the use of solar energy, CO_2_, water (H_2_O), and nutrients from the soil and the atmosphere, herein referred to as environmental factors, which in fact are highly variable in natural and agricultural ecosystems [81,82]. The availability of plant nutrients in a usable form and in sufficient amounts and appropriate ratios or balance is critical for plants to complete their life cycle [82].

Each of the essential nutrients has a definite and specific function to perform in the growth and development of plants, and a deficiency of any of them causes abnormal or restricted growth. Table 1 summarizes the main functions of N, as well as the effects caused by its deficiency and toxicity or excess versus those of other macronutrients (adapted from [83]). Appendix A contains similar details for secondary macronutrients and micronutrients.

### 6.1. Nitrogen Is a Major Constituent of Living Organisms

Nitrogen (N) and phosphorus (P) are crucial components of the DNA (deoxyribonucleic acid) and RNA (ribonucleic acid). DNA and RNA, molecules that carry genetic information for the development and functioning of an organism, are made of nucleotides including a five-carbon sugar backbone and phosphate groups (together forming the 5′–3′ phosphodiester linkage) [84] and nitrogenous bases (organic molecules containing N that act as a base in chemical reactions: adenine (A: C_5_H_5_N_5_), guanine (G: C_5_H_5_N_5_O) (purines), cytosine (C: C_4_H_5_N_3_O), and thymine (T: C_5_H_6_N_2_O_2_) or uracil (U: C_4_H_4_N_2_O_2_) in RNA (pyrimidine). The nucleobases are recognized as building blocks of DNA and RNA. The pyrimidine and purine biosynthesis pathways have been widely investigated [85,86,87], and a few genes encoding proteins controlling these pathways have been functionally characterized [16,88,89,90]. A recent study by Rolly and Yun [91] reported that an interplay exists between the de novo pyrimidine biosynthesis pathway and N metabolism in plants. In addition, N interacts with other elements, such as carbon [92,93,94] and potassium [95], to maintain a nutrient balance in plants under various conditions.

### 6.2. Nitrogen Is a Key Element in Plant Physiology and Biology

It is widely accepted that N is indispensable for plants in many ways (Figure 2, [96]). Stein and Klotz [97] indicated that N is the fourth most abundant element in cellular biomass. The authors also indicated that the interchange between inert N_2_ in the extant atmosphere and reactive nitrogen compounds that support or are products of cellular metabolism and growth is controlled by microbial activities. Nitrogen also plays an important role in various physiological, biological, molecular, and biochemical processes in plants under various environmental conditions. Bassi et al. [98] supported that N is a major component of the photosynthesis apparatus. The authors observed that the application of N during sugarcane cultivation influenced photosynthesis establishment. Similarly, Evans [99] indicated that there is an established relationship between photosynthesis and N in leaves of C3 plants, and that the photosynthetic capacity of leaves is related to the N content, whereby the proteins of the Calvin cycle and thylakoids represent the majority of leaf N. Similar studies reported the role of N in photosynthesis [100,101,102,103,104], while a balanced cell-cycle control is required to ensure the balance with adaptation to dynamic environmental conditions [105].

A study conducted by MacAdam et al. [106] involving the application of N to tall fescue plants increased leaf elongation rate (LER), as a result of epidermal cell elongation and mesophyll cell division. On the other hand, N starvation was shown to alter cell division but not initiation of rice tiller buds in rice [107]. Lin and Tsay [108] demonstrated that different levels of NO_3_ and N supply differentially influenced the control of flowering in *Arabidopsis*. From the same perspective, Moreno et al. [109] suggested that NO_3_ defines shoot size via compensatory roles for endoreplication and cell division in *Arabidopsis*.

Recent studies have shown that N is involved in plant signaling and gene expression both under normal conditions and in response to abiotic and biotic stress conditions [110,111,112,113,114,115]. Moreover, Fritz et al. [116] reported the carbon-nitrate status in tobacco regulate secondary metabolism, with NO_3_ inhibiting phenylpropanoid metabolism. In a converse approach, Ibrahim et al. [117] reported that the application of N-rich fertilizer affected the synthesis of primary and secondary metabolites in *Labisia pumilia* Blume (Kacip Fatimah). Oksman-Caldentey et al. [118] recorded a change in primary and secondary metabolism in roots of *Hyoscyamus muticus* caused by N and sucrose supplementation.

### 6.3. Nitrogen Is Required for Hormonal-Mediated Signaling in Plants

The multiple roles of phytohormones in physiological processes and signaling during abiotic or biotic stress events have been established, and progress continues to be made regarding their crosstalk, as well as their interactions with other molecules or cellular compounds [119,120,121,122,123,124]. A study conducted by Kiba et al. [125] highlighted that plants consist of multiple organs with various functions and different nutritional requirements, and they strongly rely on local and long-distance signaling pathways. In this regard, phytohormones, such as auxin (indole-3-acetic acid, IAA), abscisic acid (ABA), or cytokinin (CK) have been shown to interact with N signaling to coordinate the responses to fluctuations in N supply at the whole plant level. Similarly, Xu et al. [126] reported an interaction between auxin, CK, and strigolactone (SL) and N availability in the regulation of shoot branching in rice.

In addition, Vega et al. [127] revealed the role of NO_3_ in the modulation of hormonal pathways, while suggesting that plant hormones participate in local and systemic N signaling to regulate root and shoot growth, and that hormone crosstalk influences NO_3_ uptake. Likewise, Wen et al. [128] revealed that phytohormones interact with NO_3_ to regulate plant senescence. In the same way, Wang et al. [129] found that gibberellins (GAs) are involved in the regulation of N uptake and other physiological traits in maize in response to N availability. It was also shown that brassinosteroids modulate N physiological response and promote N uptake in maize [130].

### 6.4. Genes Involved in the Regulation of Nitrogen Uptake, Transport, and Assimilation in Plants

Genes involved in the N metabolism have been widely investigated and characterized under various environmental conditions [131,132,133]. Their interplay helps plants to maintain a balanced N availability and use efficiency. To date, five families of NO_3_ transporters have been reported, grouped into low- and high-affinity [134,135], including NO_3_ transporter 1 (NRT1), NO_3_ transporter 2 (NRT2), chloride channel (CLC), and slow anion channel-associated/slow anion channel-associated homologs (SLAC/SLAH) [136,137,138,139], with the NRT1 protein family harboring the biggest number [91]. Their activity is induced when plants experience low or high N availability in the soil. Among the well-characterized NO_3_ transporters, in various plant species, the *Arabidopsis* NRT2.1, a pure high-affinity NO_3_ transporter, was reported to be suppressed by high NO_3_ levels, but activated under low NO_3_ conditions [140]. In contrast, the NRT1.1 family acts exceptionally as dual-affinity NO_3_ transporter, which switches from low-affinity under regular NO_3_ availability to high-affinity under low NO_3_ availability. The fluctuation from low to high affinity of NRT1.1 is said to be facilitated by phosphorylation at the Thr101 residue of NRT1.1 that enhances its affinity to NO_3_ [141,142]. As illustrated in Figure 3, most of the NO_3_ or NH_4_ taken up by the plant is transported and translocated to the shoots where it is reduced to nitrite (NO_2_) via the activity of nicotine amide dinucleotide phosphate (NADPH)-dependent cytosolic NR. A certain amount of NO_2_ is then translocated to the chloroplasts where it is reduced to NH_4_ for further assimilation into organic compounds, such as amino acids. This event occurs through the action of the glutamine/glutamate synthase and glutamine-2oxoglutarate aminotransferase system [143,144].

The advances in plant molecular research and the advent of sequencing technologies and bioinformatics have allowed the discovery of several members of the low- and high-affinity NO_3_ and NH_4_ uptake, transport, translocation, assimilation, and remobilization molecular functions in several plant species in response to various environmental stimuli such as drought stress [138,146,147,148,149,150,151,152,153,154], salinity [155,156,157,158,159], heat stress [160,161], heavy-metal stress [162,163], flooding stress [164,165,166], and biotic stress [167,168,169,170,171].

## 7. Multiple-Nutrient Deficiency and Toxicity

Plants absorb N in specific chemical forms, and some of these forms are naturally present in soils (Appendix A). Plants need the most N during their vegetative and development stages. A balanced N supply can result in optimum growth and development. Although total nutrient content in the soil might be abundant, many nutrients are poorly available for plant uptake, in part due to a series of physical and chemical reactions, as well as biological processes, which exert an influence on the form on which they exist in the soil [68,82,95,172,173]. Nitrogen is available to the plant in the form of NO_3_ and NH_4_, with NO_3_ being the major form. Nitrate is negatively charged and, therefore, cannot bind to soil particles that are also negatively charged. In addition, plants usually take up about half of the N available in the soil. The other fraction of N is lost through either evaporation, runoff, or leaching in the depths of the soil to the aquifer (groundwater).

In addition, nutrient availability may be hindered by several external factors to the plant, such as the potential of hydrogen (pH, alkaline, neutral, or basic) [174,175]. The pH, a measure of the acidity or alkalinity, is a major component of nutrient acquisition by plants from the soil [176]. Soil pH level has an influence on the availability of nutrients to plants, which may occur through the modification of the basic form of a particular nutrient element in the soil. A soil pH of 6.0–7.5 is acceptable for most nutrient elements, but the optimum varies among plant or crop species [177]. Soil pH also determines microbial diversity and composition, including those mediating the decomposition of OM and the conversion of chemical elements from non-available forms to available forms for plants [178,179,180].

Likewise, the abundance of salts in the soil [181,182] may affect the mobility of key chemical components in the soil. Other reasons include the structure and texture of the soil (porosity of soil, cationic exchange capacity (CEC), etc.) [183], the OM in the soil, the activity of microorganisms in the soil that may be competing with the plant for their own food for survival, the availability of water in the growing environment (soil), which may be compromised by the water retention capacity of the soil (porosity and texture), and the heavy-metal contamination of soils. This implies that nutrients may be present in the soil but still unavailable to the plants.

Nitrogen deficiency also occurs when carbon-rich OM such as sawdust is supplemented to the soil. N and carbon metabolisms are functionally linked, and a balanced carbon/nitrogen (C/N) ratio is of great importance in ensuring N availability to the plant [92]. C/N also determines the level of the activity of telluric microorganisms and their multiplication. During the initial stage of fresh OM decomposition, soil microorganisms rely on the N contained in the OM to decompose the high carbon content in the OM. These microorganisms can also use any N available in the soil to break down carbon sources, thus making N unavailable to plants. This phenomenon is known as “robbing” the soil of N.

Moreover, soil erosions and runoff are also known to lead to N loss from agricultural lands [184]. It has been well established that N-deficient soil initially affects plant growth and development, causing the plant to be thin, pale, and subject to chlorosis, with low protein levels in grains, low biomass, and eventual crop failure [185,186,187,188]. Therefore, early detection of N deficiency may help to make decisions quickly so as to prevent crop failure and economic losses [189,190,191]. In addition to these factors, abiotic stress such as drought, salinity, and heavy-metal toxicity can be the cause of nutrient (including N) unavailability to the plant. As a result, the leaves take on a burnt look from dehydration and nutrient deficiency. Under these conditions, water availability to the plant may be hindered, and nutrient uptake may be impaired.

In contrast, excessive N supply is detrimental to the plant [192]. Although different plant species require different levels of N for peak health, with various thresholds for N toxicity, the symptoms of excessive N application favors foliage growth (overly or dark green) or vegetation growth and elongation of the plant (reduced stem strength), which may destabilize the plant’s stability in its soil position (sensitive to lodging during flowering and grain filling). Likewise, excessive N supply may stress the roots, causing root growth stunting, as well as delayed flowing and grain formation, consequently affecting grain yield and eventually death of the plant due to the stress. The use of excessive N concentrations in soils may result in N toxicity (commonly observed through the discoloration of leaves (from edges of the leaf and spread inward). Too much N in the soil also increases soil mineral salts and luxuriant growth, resulting in the plant being attractive to insects and/or diseases/pathogens (increased susceptibility to bacterial and fungal diseases). Additionally, excessive N levels in soils have negative implications for the environment, such as groundwater pollution due to the leaching of excessive N that the plant cannot absorb through water runoff or percolation (which may contaminate drinking water and groundwater). Other effects of excessive N on the environment include an enhanced production of GHGs, such as CH_4_ and N_2_O, which are later released to the atmosphere through gas exchange (90% mediated by plants) or diffusion from soil (10%). Nevertheless, knowing that different plant species require different levels of N for peak health, the thresholds for N toxicity also differ from one plant species or crop to another.

## 8. Tracking Nitrogen Uptake and Assimilation in Plants

Fertilizers labeled with radioactive isotopes, such as phosphorus and nitrogen-15 have been used to investigate fertilizer uptake, retention, and utilization [193,194,195]. The N-15 (^15^N) isotopic technique, which may help identify the source of N_2_O generation and reduce the emission of this potent GHG during nitrification and denitrification processes, offers comparative advantages over conventional techniques for measuring the impact of climate change [195]. Stable isotope enrichment approaches have long been established to trace the source of N_2_O following the application of ^15^N-labeled fertilizers, such as ^15^N-labeled NH_4_ and NO_3_ [196]. This technique enables the determination of the source of fertilizer-derived ^15^N-N_2_O.

In general, nitrification derived N_2_O is quantified upon the supply of ^15^NH_4_, while that mediated by denitrification is measured following the supply of ^15^NO_3_. In the same way, considering that multiple pathways mediating N_2_O formation and consumption occur simultaneously in various microenvironments in soils, nitrification inhibitors and isotope signature techniques are commonly utilized to separate N_2_O-producing and -reducing pathways [197]. Stevens and Laughlin [198] suggested that the reduction of N_2_O to N_2_ could be quantified by determining ^15^N in N_2_ following the supply of ^15^NO_3_. Similarly, Baggs et al. [199] demonstrated that application of ^14^NH_4_/^15^NO_3_ and ^15^NH_4_/^14^NO_3_ helped determine the relative contributions of nitrification and denitrification to N_2_O production. Furthermore, He et al. [200] proposed the use of the stable ^15^N isotope as a means for quantifying N transfer between mycorrhizal plants, considering that plants acquire nutrients from soil, and mycorrhizae play vital roles in plant nutrient acquisition, performance, and productivity. In the same way, the use of the carbon-13 (^13^C) stable isotopic technique helps evaluate the source of carbon sequestered in the soil.

Isotopic fractionation [201,202,203,204] can cause the isotope amount ratio n(^15^N)/n(^14^N) to increase systematically through food chains via assimilation of N compounds in biomolecules such as proteins. Isotopic fractionation occurs because of assimilation, storage, and excretion of proteins and other N compounds. Isotope amount ratio n(^15^N)/n(^14^N) measurements have been widely used to test hypotheses about predator–prey relations and detect disruptions to the trophic structure of ecosystems that might be caused by toxic contaminants, invasive species, harvesting, or organisms. Similar principles are used to detect differences in diets among animals, including humans [205,206,207].

Artificially enriched ^15^N tracers are used to study movement and transformation of N in biological and environmental systems, such as the uptake and loss of N fertilizers by crops. A common experiment involves introducing an isotopically labeled compound into the environment and then analyzing various samples taken from the environment for the presence of the enriched isotope to determine where the labeled compound moved and whether it transformed into another compound. Artificially enriched ^15^N has also been employed to study uptake and dispersal of N in feed supplies used in food production industries such as aquaculture [208].

The stable isotopes of N are subject to isotopic fractionation via physical, chemical, and biological processes. Variations in the isotope amount ratio n(^15^N)/n(^14^N) are commonly used to study Earth system processes, especially those related to biology, because N is a major nutrient for growth [209]. To illustrate this, isotope fractionation occurs when dissolved solutes, such as nitrate (NO_3_), are transformed to more reduced compounds (i.e., N_2_) because NO_3_ with higher ^14^N abundance tends to be more readily broken down. This leaves the residual unreacted NO_3_ with a higher n(^15^N)/n(^14^N) ratio than the initial ratio prior to reaction. Changes in the isotopic composition of biologically reactive compounds can be used to detect such reactions in aquatic environments, which are important mechanisms for removing reactive contaminants such as NO_3_ [210].

Variation in N stable isotopes has been used to track fertilizer N accumulation into plants, soils, and infiltrating groundwater to improve efficiency and reduce impact on the environment, such as experimental agricultural fields where various amounts of excess N from fertilizers and plant residues can be found in groundwater.

## 9. Paradigm Shift to a Sustainable Agricultural Production System

Agriculture is a major source of food. Food production is required to double within the next 8–10 decades or so from the perspective of the rapid population growth. Agriculture has been identified as a sink and source of GHGs, as well as the economic sector that suffers the most from climate change. As a source, reports support that the application of N-rich fertilizers is regarded as one of the factors contributing significantly to enhancing GHGs formation during crops cultivation. Rodale [211] proposed paths to transition to a sustainable agricultural production system on the basis of the following observations: (i) because renewable resources are the basis of operation and productivity of modern agriculture, in the event of depletion of nonrenewable resources, either food will become extremely expensive or productivity will decline; (ii) the current food production system contributes to the environmental degradation (soil erosion, soil degradation, and deforestation); (iii) lines of evidence indicate that a number of agricultural practices, including the pattern of N-rich fertilizer application, contribute to the escalation of pollution problems; (iv) there is a strong agreement that the natural resources are limited and should be used in a sustainable manner; (v) conventional technologies and secular agricultural production systems are likely to be unsustainable in the future in the event that agricultural production becomes the major source of energy and feed stocks; (vi) a major concern exists over whether the good life in rural areas can be maintained if family farms are replaced by large-scale industrialized farms, which produce all the food.

Owing to the above, sustainable agriculture, as proposed by diverse sources and condensed by the International Food and Agriculture Development (IFAD) task force report [212], could be portrayed as follows: (i) the successful management of resources to satisfy changing human needs, while maintaining or enhancing the natural resources base and avoiding environmental degradation; (ii) the ability of an agricultural system to maintain production over time in the face of social and economic pressures; (iii) one that should conserve and protect natural resources and allow for long-term economic growth by managing all exploited resources for sustainable yields. Other sources argue that stainability can only be achieved when resources, inputs, and technologies are within the capabilities of the farmers to own, hire, and manage with increasing efficiency to achieve desirable levels of productivity with minimal effects on the resources base, human life, and environmental quality. In this regard, sustainable agricultural production system is referred to as “one that maintains an acceptable and increasing level of productivity that satisfies prevailing needs and is continuously adapted to meet the future needs for increasing the carrying of the resource base and other worthwhile human needs” [213,214,215,216].

Nevertheless, the major challenge remains the reduction in GHG emissions from agriculture during crop cultivation. The last two decades have been marked by a hunt for sustainable and effective strategies toward achieving a sustainable agriculture, here referring to agricultural practices that help reduce, in a sustainable manner, the emission of GHGs. Among them, improving N use efficiency (NUE, from uptake to assimilation and remobilization) is considered the most promising and effective approach, because it allows significantly reducing the application of N-rich fertilizers, which helps cut down the production cost, with a low impact on the environment. Reports have demonstrated that NUE is controlled genetically, and this complex trait allows optimizing the use of N available to the plant. NUE is equally important when plants experience nutrients shortage or unavailability caused by environmental stresses, such as drought [153,217,218], salinity [219], heat stress [220], or heavy-metal toxicity, whereby the acquisition of N is either restricted or impaired. In addition to the well-characterized NO_3_ or NH_4_ transport- and assimilation-related genes [221], N remobilization (one of the components of NUE) from NO_3_ stored in the vacuole regulated through a process known as autophagy carries the potential to salvage poor N supply or transport within the plant, and it offers an alternative for balanced plant productivity and reduced GHG emissions [220,222,223,224]. Likewise, NO, an ancient molecule with multiple roles in the plant, previously suggested to play an important role in N acquisition and assimilation events through NR activity [225,226], could serve as a potential target for improving NUE in plants.

Other alternative approaches may include intermittent drainage, especially in flooded or irrigated cultivation systems, the use of nitrification inhibitors, and the application of biochar. Unlike in the industrial sector, it is important to indicate that net zero GHG emission may not be a realistic target from agriculture with regard to the biological nature of CH_4_ and N_2_O production, involving soil microorganisms such as methanotrophs and methanogens, and the requirement of CO_2_ in plant metabolic processes.

## 10. Conclusions and Future Perspectives

Nitrogen (N) may be to the plant, to some extent, what the seed is to agriculture. Efficient plant growth, development, and productivity are conditioned by the quality of nutrients available, coupled with the ability of the plant to take up and use them. On the one hand, N is one of the most important macronutrients, and by far the most abundantly used in agricultural production. Today, conceiving agriculture without N is nearly utopic, considering that N is vital for the plant and is involved in almost all physiological and biological processes. It is believed that, in order to reduce the impact of current agricultural practices on the environment, optimizing applications of N-rich fertilizers, enhancing the N use efficiency, and implementing better crop management practices are key factors to alleviate the impact of climate change, while maintaining a balanced productivity and quality.

## Figures and Tables

**Figure 1 life-12-01272-f001:**
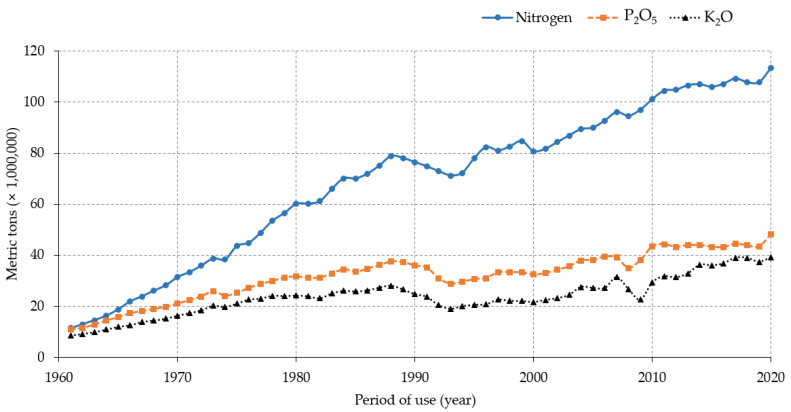
Pattern of global macronutrients use in agriculture from 1961 to 2020.

**Figure 2 life-12-01272-f002:**
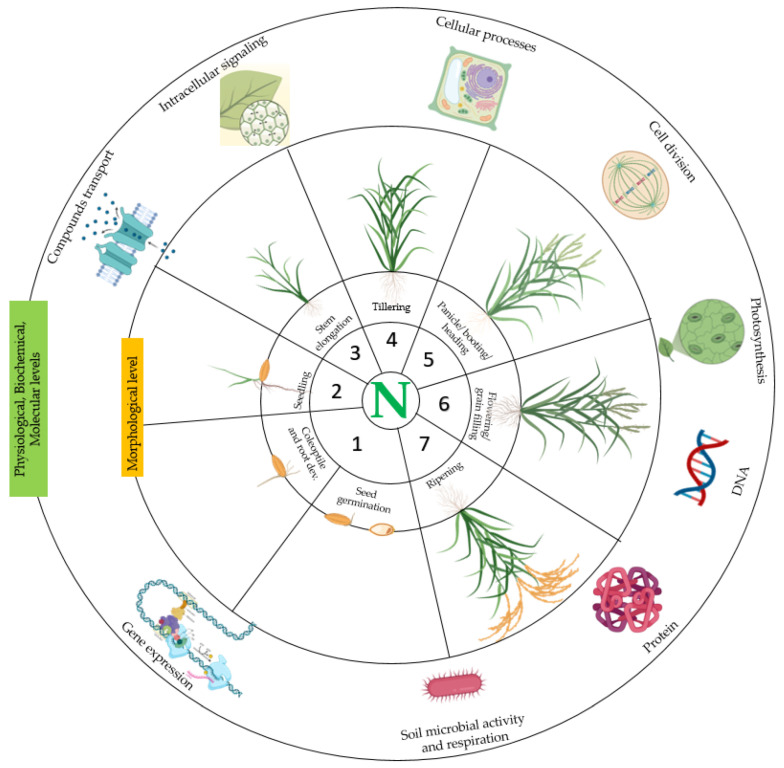
Illustration of the multiple roles or involvement of nitrogen in plant growth, development, and cellular metabolism. Nitrogen is contained in various biological components and takes part in diverse physiological processes, biochemical reactions, and molecular response toward stress in plants. This illustration was created using the gallery proposed by Biorender (https://app.biorender.com/, accessed on 23 March 2022). The circular shapes and the description were added using Microsoft Office PowerPoint 2016.

**Figure 3 life-12-01272-f003:**
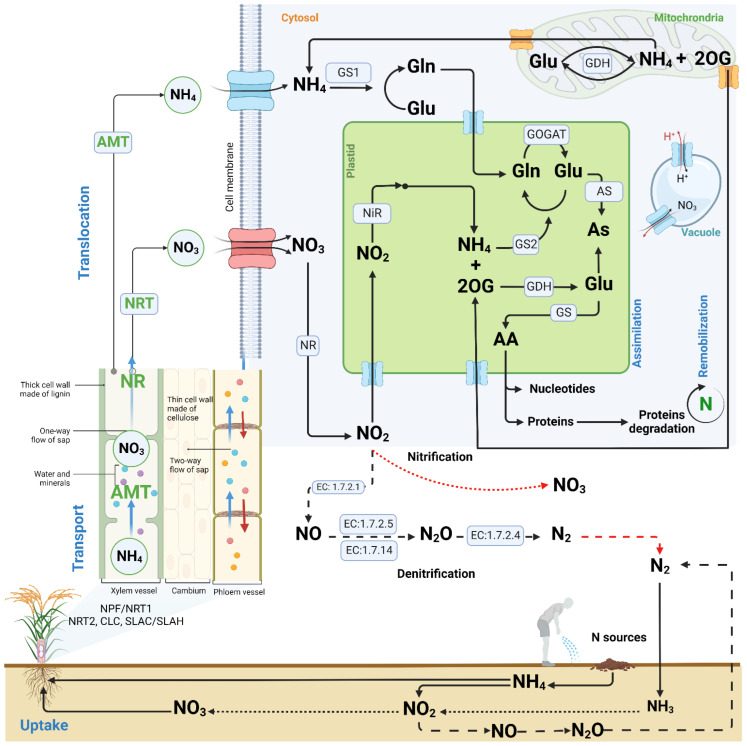
Illustration of N metabolism in plants. N is acquired by plants from the soil, which is the major source. Soil microorganisms are responsible for mediating the conversion of the atmospheric N gas (N_2_) to a form available to the plant (NO_3_ or NH_4_). N is also supplied exogenously through fertilizer application. NO_3_ and NH_4_ are taken up by the roots and transported to the shoot via the action of several transporter-encoding genes, which translocate to the cells. NO_3_ and NH_4_ undergo series of conversions and are assimilated in the form of amino acids, becoming part of several macromolecules. The NO_3_ stored in the vacuole is remobilized in the event of reduced fluxes from soil to the plant. This model was created from available information on the nitrogen metabolism proposed in the literature, including Wang et al. [131], Ali [145], and the Kyoto Encyclopedia of Genes and Genomes (KEGG) pathways database (https://www.genome.jp/pathway/ko00910, accessed on 10 June 2022) using the BioRender design platform (https://app.biorender.com/, accessed on 10 June 2022). Continuous lines with an arrow indicate a positive process or induction. Dotted and dashed lines denote the nitrification and denitrification processes, respectively.

**Table 1 life-12-01272-t001:** Beneficial outcomes and advert effects of nitrogen versus phosphorus and potassium.

Plant Nutrients	Function	Deficiency Symptoms	Excess/Toxicity
Primary nutrients (NH_4_^+^/NO_3_^–^, H_2_PO_4_^–^/HPO_4_^–^, and K^+^)			
Nitrogen (N)	- An important constituent (or involved in the synthesis) of chlorophyll, protoplasm, proteins, coenzymes, and nucleic acids. - Increases growth and development of all living tissues. - Improves the quality of leafy vegetables and fodders and the protein content of food grains.	- Stunted growth. - Appearance of a light-green to pale-yellow color on older leaves, starting from the tips. Lower leaves turn yellow. This is followed by death and/or dropping of the older leaves depending upon the degree of deficiency. - In acute deficiency, flowering is greatly reduced. - Lower protein content.	- Deep-green vegetation and poor secondary shoot growth development. - Greater bud formation instead of reproductive bud formation. - Roots turn brown, with necrotic root tips. - Reduced plant growth. - Necrotic lesions occur on stem and leaves. - Vascular browning occurs in stems and roots. - Increases soil mineral salts. - Promotes luxuriant growth, resulting in the plant being attractive to insects and/or diseases/pathogens (increases susceptibility to bacterial and fungal diseases). - Enhances production of methane (CH_4_) and nitrous oxide (N_2_O).
Phosphorus (P)	- A constituent of phosphatides, nucleic acids, proteins, phospholipids, and coenzymes, metabolic transfer processes, NAD, NADP, and ATP, photosynthesis, and respiration. - Constituent of certain amino acids. - Necessary for cell division, a constituent of chromosomes, and stimulates root development. - Necessary for meristematic growth, seed and fruit development, and stimulates flowering.	- Overall stunted appearance, where the mature leaves have characteristic dark to blue-green coloration, and restricted root development. - In acute deficiency, occasional purpling of leaves, especially the margins, and the stem shows spindly growth. - Delayed maturity and lack of or poor seed and fruit development. - Reduced tillering in cereals. - Yield loss ranging 10–15% of maximal yield.	- Excess P in the plant can result in iron (Fe) and Zinc deficiencies.
Potassium (K)	- Involved in sugar and starch formation, lipid metabolism, and nitrogen fixation, and neutralizes organic acids. - An activator of enzymes involved in photosynthesis and protein and carbohydrate metabolism. - Assists carbohydrate translocation, synthesis of protein and maintenance of its stability, membrane permeability and pH control, and water utilization by stomatal regulation. - Improves utilization of light during cool and cloudy weather, thereby enhancing plant ability to resist cold and other adverse conditions. - Enhances the plant’s ability to resist diseases. - Increases the size of grains or seeds and improves the quality of fruits and vegetables. - Plays a role in osmoregulation of water use (opening and closing of stomata).	- Chlorosis along the leaf margins (or marginal burning of leaves and curling of leaves) followed by scorching and browning of tips of older leaves; these symptoms then gradually progress inward. - Slow and stunted growth of plants. - Stalks weak, and plants lodge easily. - Shriveled seeds or fruits.	- High amounts of K can cause calcium, magnesium, and nitrogen deficiencies.

## Data Availability

Not Applicable.

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
