# Peer review of "Multiple Facets of Nitrogen: From Atmospheric Gas to Indispensable Agricultural Input"

_life, 2022, doi:10.3390/life12081272_

Round 1

Reviewer 1 Report

General comments

Please check the MDPI format on references!!

Please revise the article based on the MDPI format!!!

Please answer my questions:

What is the objective of this study?

What is the novelty of this study?( it is no clear for me)

Specific comments

Line 16 The role of N change to N's role

Line 54-58 English revise

Line 114-115 needs to revise

Line 133 are change to have

Line 180-181 needs to revise

Line 207 needs to write references number

Line 239-241 not clear

Line 295-298 needs to revise

Line 323-327 needs to English revise

Line 388-389 needs to revise

Line 466-467 needs to revise

After major revision can be accept.

Author Response

Manuscript ID: life-1841899

Point by point reply

Reviewer 1

We are thankful to the reviewer for the time taken to review our manuscript, and most importantly for the valuable comments made towards the improvement of the content, format as well as the contribution of this review manuscript to field considered. We are happy to share that almost all concerns raised by the reviewer have been addressed, and the manuscript has been extensively revised.

General comments

Please check the MDPI format on references!!

Please revise the article based on the MDPI format!!!

We appreciate the concern raise by the reviewer. We have improved the formatting of the list of references according to the guidelines proposed by MDPI.

Please answer my questions:

What is the objective of this study?

The main objective of this review is to give readers, scientists and agricultural practitioners a comprehensive view of nitrogen in its diverse roles and multiple facets in a single spot from available reports related to nitrogen and its use in crop production. This is particularly important to put nitrogen into the perspective of the current global environmental challenges and raise awareness on the future of N, which is evidently the most abundantly used mineral elements in agricultural production.

What is the novelty of this study?( it is no clear for me)

We appreciated the concern raised by the reviewer.  We would like to share that this review, basically assessed the current knowledge of nitrogen and N-containing compounds in their multiple facets. Nitrogen is an important element abundantly used in agriculture, and today it is nearly utopic to conceive doing agriculture without nitrogen. Unlike other review reports on nitrogen, this work proposes a panoramic view of nitrogen, tracking the use of nitrogen in agriculture from empirical background to the current era. Since the domestication of plant species, our understanding of N has increased significantly, particularly with the advent of sequencing technologies and bioinformatics. The role of N is scattered in the literature, which makes difficult to draw a big picture of the dynamics of N in plant metabolism, and its usefulness in plants fitness, as well as its contribution to enhancing production of greenhouse gases. The current, global warming majorly caused by GHG imposes an imperative to rethink the use of N-sources during crops cultivation towards a sustainable agriculture.

Less attention was given to other aspects of N, not because they were not reported, but in part because they were not connected to give a comprehensive view. This review also proposes that nitrogen use efficiency has a dual benefit of improving the use of N by plants, while reducing external supply of N-rich fertilizers, and lowering the impact on the environment.   

To facilitate the visualization of different aspects of N in the literature, we have constructed various illustrations that allow to see different pathways showing the mechanism of N acquisition, transport, translocation, assimilation, and remobilization.

We have included the possible forms of N commercially available versus the form that plants are able to take up.

We expect that this review help the scientific community, the industry, and the public to rediscover nitrogen, while putting N in a wider environmental perspective. This could trigger novel studies and initiatives for a better use of N considering that N is a limiting factor in agriculture and environmental aspects when interacting with other elements or compounds.

Specific comments

We would like to thank for the time taken to review this manuscript and for having suggested valuable comments for improvement.

Line 16 The role of N change to N's role

We made the necessary change as suggested

Line 54-58 English revise

We have tried to improve the statement as follows: Likewise, abiotic factors, such as salinity, drought stress, or heavy metal toxicity can impair the availability of nutrients to the plant, thus resulting in poor growth and productivity (Lines 53–55)

Line 114-115 needs to revise; Line 133 are change to have

We have extensively revised the manuscript, including the paragraphs highlighted by the reviewer. Some texts have been moved to other sections to improve the flow of information and keep some logic in the description.

Line 180-181 needs to revise

Line 207 needs to write references number

Line 239-241 not clear

Line 295-298 needs to revise

We have revised as suggested. The text is moved to lines 202–204.

Line 323-327 needs to English revise

We have improved the paragraph, which is now moved to line 262.

Line 388-389 needs to revise

We revised the lines 374–376 (previously 388–389) as suggested.

Line 466-467 needs to revise

We modified the statement in question for a better readability (see lines 470–472)

Reviewer 2 Report

This review aimed the multiple facets of N and N-containing compounds in plants, some of the ancient and advanced knowledge regarding N use, availability, deficiencies, and side effects. The manuscript aimed to raise awareness the future use of N  in crop production, too.

Despite its purpose, the review lost its centrality on nitrogen, shifting the focus to other nutrients, in some paragraphs such as paragraphs 3, 4, 5, table 1, 2.

Paragraph 6 had to propose solutions to reduce the use of nitrogen inputs into the environment, but this is missed.

In my opinion, the work cannot be accepted in the current form.

Best regards.

Author Response

Manuscript ID: life-1841899

Point by point reply

Reviewer 2

This review aimed the multiple facets of N and N-containing compounds in plants, some of the ancient and advanced knowledge regarding N use, availability, deficiencies, and side effects. The manuscript aimed to raise awareness the future use of N in crop production, too.

We sincerely appreciate the comments made by the reviewer. We have revised extensively the manuscript as suggested by the reviewer, and we are happy to share that almost all concerns raised by the reviewer have been addressed, and

Despite its purpose, the review lost its centrality on nitrogen, shifting the focus to other nutrients, in some paragraphs such as paragraphs 3, 4, 5, table 1, 2.

We appreciate the concerns raised by the reviewer. We would like to apologize for any inconvenience caused by any missing points or excessive information provided in this review manuscript. We have deeply revised the manuscript, including the paragraphs highlighted and tables.

While taking positively these comments, we would like to highlight the following:

-          Nitrogen is the central element and the target of this review. In the perspective of giving a brief historical background, we thought of going back in time to trace the use of N in agriculture. This analysis took us to the domestication of plant species and the late development of mineral nutrients for plants, including nitrogen as a major macronutrients.

-          To further corroborate the views from various sources indicating the indispensable character of nitrogen for plant productivity and quality, we performed a data mining to investigate the trend in the use of nitrogen versus other macronutrients (phosphorus and potassium) over a period of 62 years (from 1960–2022). This gives a big picture and is a form of evidence to support the essentiality of N relative to other nutrients as displayed in Figure 1.

-          Tables 1 and 2 have been moved to supplementary materials

Paragraph 6 had to propose solutions to reduce the use of nitrogen inputs into the environment, but this is missed.

We would like to appreciate the quintessence of the concern raise by the reviewer. We made necessary changes in previous paragraph 6 now referred to as paragraph 9 (508–576). We hope these changes meet the reviewer’s expectations.

In my opinion, the work cannot be accepted in the current form.

Best regards.

We extensively revised the manuscript following the reviewer’s comments, and we hope that in its current form, the manuscript is readable and meet the reviewer’s expectations.

Round 2

Reviewer 1 Report

Dear Authors

I accepted your response.

good luck

Reviewer 2 Report

The manuscript was improved based on the reviewers' requests. Missing information was added, all sections were enhanced as suggested. In my opinion, the work could be accepted in the current form.

Best regards